# Effects of Dietary Supplementation with Honeybee Pollen and Its Supercritical Fluid Extract on Immune Response and Fillet’s Quality of Farmed Gilthead Seabream (*Sparus aurata*)

**DOI:** 10.3390/ani12060675

**Published:** 2022-03-08

**Authors:** Rosaria Arena, Adja Cristina Lira de Medeiros, Giulia Secci, Simone Mancini, Simona Manuguerra, Fulvia Bovera, Andrea Santulli, Giuliana Parisi, Concetta Maria Messina, Giovanni Piccolo

**Affiliations:** 1Laboratory of Marine Biochemistry and Ecotoxicology, Department of Earth and Sea Sciences-DiSTeM, University of Palermo, Via Barlotta 4, 91100 Trapani, Italy; rosaria.arena@unipa.it (R.A.); simona.manuguerra@unipa.it (S.M.); andrea.santulli@unipa.it (A.S.); concetta.messina@unipa.it (C.M.M.); 2Department of Agriculture, Food, Environment and Forestry-DAGRI, University of Firenze, Via Delle Cascine 5, 50144 Firenze, Italy; adja.cristina@gmail.com (A.C.L.d.M.); giulia.secci@unifi.it (G.S.); 3Department of Veterinary Sciences, University of Pisa, Viale Delle Piagge 2, 56124 Pisa, Italy; simone.mancini@unipi.it; 4Department of Veterinary Medicine and Animal Production, University of Napoli Federico II, Via Delpino 1, 80137 Napoli, Italy; fulvia.bovera@unina.it (F.B.); giovanni.piccolo@unina.it (G.P.); 5Institute of Marine Biology, University Consortium of the Province of Trapani, Via Barlotta 4, 91100 Trapani, Italy

**Keywords:** honeybee pollen, supercritical fluid extraction, bioactive compounds, immune system, fatty acids

## Abstract

**Simple Summary:**

In aquaculture, the aspect of nutrition is essential for fish health, quality of the products and side effects on economic profitability. In recent years, the interest on natural compounds, as fish health promoters, is exponentially increasing and this is going to support the development of new and balanced commercial diets for aquaculture. Honeybee pollen (HBP) is a good source of bioactive compounds that have demonstrated to have significant beneficial effects on immunity responses in farmed species. The aim of this study was to investigate the effects of HBP and its extract on some parameters related to the immune response and quality, in farmed gilthead seabream (*Sparus aurata*). Isoenergetic and isoproteic diets were formulated as follows: a control diet (CTR); two diets containing 5% (P5) and 10% (P10) of HBP inclusion; two diets containing HBP extract, obtained by supercritical fluid extraction (SFE) (HBP_SFE), at 0.5% (E0.5) and at 1% (E1). The results demonstrated that the diet E1 improved both fish health-related parameters while not impairing fillets quality, suggesting that this level of supplementation could be suitable for gilthead seabream inflammatory responses.

**Abstract:**

The awareness of the correlation between administered diet, fish health and products’ quality has led to the increase in the research for innovative and functional feed ingredients. Herein, a plant-derived product rich in bioactive compounds, such as honeybee pollen (HBP), was included as raw (HBP) and as Supercritical Fluid Extracted (SFE) pollen (HBP_SFE) in the diet for gilthead seabream (*Sparus aurata*). The experiment was carried out on 90 fish with an average body weight of 294.7 ± 12.8 g, divided into five groups, according to the administration of five diets for 30 days: control diet (CTR); two diets containing HBP at 5% (P5) and at 10% (P10) level of inclusion; two diets containing HBP_SFE, at 0.5% (E0.5) and at 1% (E1) level of inclusion. Their effects were evaluated on 60 specimens (336.2 ± 11.4 g average final body weight) considering the fish growth, the expression of some hepatic genes involved in the inflammatory response (*il-1β, il-6* and *il-8*) through quantitative real-time PCR, and physico-chemical characterization (namely color, texture, water holding capacity, fatty acid profile and lipid peroxidation) of the fish fillets monitored at the beginning (day 0) and after 110 days of storage at −20 °C. The results obtained showed that the treatment with diet E1 determined the up-regulation of *il-1β*, *il-6*, and *il-8* (*p* < 0.05); however, this supplementation did not significantly contribute to limiting the oxidative stress. Nevertheless, no detrimental effect on color and the other physical characteristics was observed. These results suggest that a low level of HBP_SFE could be potentially utilized in aquaculture as an immunostimulant more than an antioxidant, but further investigation is necessary.

## 1. Introduction

It is well known that in aquaculture, an optimal nutritional status is essential for fish welfare, quality and economic production [1], and for these reasons the interest in new and balanced commercial diets has grown significantly in recent years [1,2]. Supplementation with natural compounds has proved to be a useful tool to improve health production in the aquaculture industry [3] by minimizing the risk and side effects of synthetic products [4] and obtaining safe products intended for human consumption [5]. In particular, special emphasis has been given to bioactive molecules, plant or medicinal herb derived-products which could improve health status, innate and adaptive immune responses as well as increase the disease resistance in various fish species [6,7,8]. Furthermore, while acting as anti-stress and anti-infection agents, the use of some plant products in fish diets can promote other positive effects, such as improving growth, weight gain, appetite stimulation, and the early maturation of cultured species [8,9].

Dietary intervention with natural extracts, essential oils, and plant by-products was studied even to enhance meat quality and stability during storage, according to the consumer’s demand for “natural animal feeding”. A valuable example was recently proposed by Santos and colleagues [10], who found out that an essential oil blend, administrated to rainbow trout, asserted specific protection against lipid and protein oxidation of fish fillets stored at –10 °C for 6 months [10]. Besides, gilthead sea bream (*Sparus aurata*) shelf life was increased by supplementing the diet with thymol due to the reduction of spoilage and lipid oxidation damages [11].

Among others, honeybee pollen (HBP) is a plant-derived product considered an important functional food for human consumption due to its potential source of energy. In fact, HBP is defined as the “only perfectly complete food” [12], as it contains amino acids, antioxidants, minerals, vitamins and lipids [8,13,14,15], which promote health. In addition, HBP has a high antioxidant capacity [8,12,15], due to the presence of phenolic compounds [8,15,16] that are able to inactivate free radicals or to prevent their production [17], being useful in the prevention of oxidative-stress related diseases [12]. Recently, the beneficial effect of dietary supplementation with bee pollen was observed in the immune response of zebrafish [18], and also in the quality of rainbow trout fillet [19].

It has been observed that zebrafish fed with a bee pollen-added diet showed changes in the expression of immunity-related genes with improvement in the immune response [18], and characteristics of fillet [19]. Other studies tested bee pollen in animal diets also demonstrating biological effects and nutritional functions positive for growth performance [19,20,21,22,23], immunity [8,22,23,24], intestinal function [23,25,26], reproductive traits [20,22] and/or meat fish color and quality [19,23]. However, negative effects of dietary HBP were highlighted in meagre (*Argyrosomus regius*) [15], probably because pollen can be a source of various anti-nutritional compounds and indigestible fractions. These components could be possibly eliminated using supercritical fluid extraction (SFE) [13].

Supercritical fluids are highly compressed fluids that combine the properties of gases and liquids; they possess better transport properties than liquids by diffusing more easily through solid materials, which accelerate the extraction yields. Carbon dioxide (CO_2_) is one of the most commonly used solvents, which has a critical temperature and pressure of 31.3 °C and 72.9 atm, respectively. Once extraction is complete and the system decompressed, the CO_2_, being a gas at room temperature, is removed and a solvent-free extract is obtained [27]. For this reason, this method is considered a green technique, as it uses less toxic organic solvents when compared to traditional extraction methods [8,27,28]. Moreover, SFE allows obtaining bioactive compounds with a high content of polyphenols and antioxidant activity offering greater selectivity [8,29].

Therefore, the aim of the present study was to evaluate the effect of the inclusion of raw HBP and Supercritical Fluid Extract (SFE) of pollen (HBP_SFE) on the immune response and quality parameters of gilthead seabream (*Sparus aurata*). Among the immune-related parameters, we focused on the expression of some hepatic genes involved in the inflammatory response, such as *il-1β*, *il-6* and *il-8*. As quality parameters, in fillets, both sensorial, nutritional and antioxidant parameters were considered, to evaluate the effects of the inclusion of the HBP on the final product characteristics. The fillet antioxidant capacity (via ABTS and DPPH) and the oxidative status (by measuring conjugated dienes and malondialdehyde equivalents) were also analyzed to complete the assessment of the primary and secondary oxidation products, respectively, related to the final product safety and security.

## 2. Materials and Methods

### 2.1. Ethical Statement and Experimental Diets

The trial was authorized by the Ministry of Health (authorization No. 651/2017-PR, issued based on Directive 2010/63/EU) and was approved by the Ethics Committee of the Federico II University of Naples (Italy). The experimental procedures of gilthead seabream were carried out at the Department of Veterinary Medicine and Animal Production—University of Naples Federico II (Naples, Italy). The results presented in this article are part of a wider series of studies on the use of HBP in fish species whose results have been already published. In particular, part of the materials and methods and the preliminary results of this trial have been reported in Messina et al. [8]. The diets were formulated to meet nutrient requirements of gilthead seabream [30,31]. Their ingredients and chemical composition have been already reported in Messina et al. [8]. Briefly, the control diet contained soybean meal (240.0 g/kg), fish meal (210.0 g/kg), corn gluten meal (190.0 g/kg), fish oil (160.0 g/kg), gelatinized starch (100.0 g/kg), wheat gluten (80.0 g/kg), mineral mix (10.0 g/kg), and vitamins mix (10.0 g/kg).

To evaluate the influence of HBP in gilthead seabream performance, the diets were isoenergetic (21.5 MJ/kg on average) and isoproteic (39.3% crude protein on average) and formulated as follows: control diet (CTR), without HBP and HBP_SFE; two diets in which HBP was included at 5% (P5) and at 10% (P10); two diets in which HBP_SFE was included at 0.5% (E0.5) and at 1% (E1). The two pollen and extract inclusion levels were chosen based on previous research and in order to have the same total polyphenol content both in P5 and P10 diets and in E0.5 and E1 diets, according to the HBP extraction yield and the extract phenol content as determined by Messina et al. [8]. The chemical composition of the experimental diets is reported in Table 1. All the diets were prepared at the laboratories of the Department of Veterinary Medicine and Animal Production, University of Naples Federico II (Naples, Italy). The HBP was finely chopped, mixed with 10 mL of water to create a paste, then mixed in fish oil and incorporated into the mixture. The HBP_SFE was provided by the Laboratory of Marine Biochemistry and Ecotoxicology, University of Palermo (Italy), under the procedure previously detailed [8]. For E0.5 and E1 diets, the extract was dissolved initially in 10 mL of fish oil, and then incorporated into the mixture. Before the final mixing, all ingredients were ground through a 0.5 mm sieve, then water was added, and the mixture was pelleted through a 3 mm dye, using a meat-grinder (Bosch mod. MFW68660, Stuttgart, Germany). The diets were dried in a ventilated oven at a temperature of 40 °C for 24 h. This temperature was chosen as it is like the temperature maintained in the hive (34.6 °C on average) to preserve the pollen quality. The feeds were stored at 4 °C until use.

### 2.2. Fish Rearing and Sampling

The gilthead seabream specimens (n = 90; 294.7 ± 12.8 g initial body weight), normally fed, were obtained from a local farm and randomly allocated to 15 fiberglass tanks with a capacity of 220 L (6 fish per tank, 3 tanks per diet). The fish were kept for 15 days under the experimental conditions, for adaptation to the new conditions. The tanks were equipped with a biological filter, mechanical sand filter and UV ray system for water sterilization. Furthermore, water quality parameters were: temperature 22 ± 0.5 °C, salinity 33 ± 2.0 g/L, dissolved oxygen 6.5 ± 1.1 mg/L, pH 7.9 ± 0.5, total ammonia nitrogen < 0.3 mg/L, nitrite < 0.01 mg/L, and nitrate < 38 mg/L and a dark:light cycle of 12:12 h. After 30 days, a total of 12 gilthead seabream per dietary treatment (336.2 ± 11.4 g average final body weight) were sampled after a 24-h fast, anesthetized by MS-222 (tricaine methane sulfonate) (250 mg/L) and dissected in cold conditions. Biopsies were collected from the liver (approximately 300 mg), immediately preserved under PUREzol Reagent (Bio-Rad, Hercules, CA, USA) at −80 °C, and subsequently analyzed at the Laboratory of Marine Biochemistry and Ecotoxicology, University of Palermo (Palermo, Italy). The fillets (n = 120) were preserved and covered by ice and sent, in polystyrene containers, to the Animal Science Section of the Department of Agriculture, Food, Environment and Forestry (DAGRI) of the University of Florence (Florence, Italy), for the analyses scheduled.

### 2.3. Gene Expression and Quantitative Real Time Determination

The total RNA was extracted from the liver by 1 mL PUREzol Reagent (Bio-Rad, Hercules, CA, USA) [32], and the concentration was assessed by spectrophotometer at 260 nm. The absorbance ratios A260/A280 and A260/A230 were used as indicators of RNA purity. Then, 1 µg of RNA was reverse transcribed for each sample, in a volume of 20 µL, by the 5X iScript Reaction Mix Kit (Bio-Rad, Hercules, CA, USA) according to manufacturer’s instructions. The amplification was performed in a total volume of 20 µL, containing 0.4 µmol/L of each primer, 1:10 (*v/v*) cDNA, 1X IQ SYBR Green Supermix (Bio-Rad, Hercules, CA, USA) and nuclease-free water. Conditions for real-time PCRs were optimized in a gradient cycler (C1000 Touch Thermal Cycler, Bio-Rad, Hercules, CA, USA) using the following run protocol: an initial activation step at 95 °C for 3 min, followed by 39 cycles of 95 °C for 10 s and 60 °C for 30 s, with a single fluorescence measurement. The melting curve program was achieved at 65–95 °C, with a heating rate of 0.5 °C/cycle and continuous fluorescence measurement. All reactions were performed in triplicate. The relative quantification of interleukin 1 beta (*il-1β*), interleukin 6 (*il-6*), interleukin 8 (*il-8*) gene expression was evaluated after normalization with the reference genes. Data processing and statistical analysis were performed using CFX Manager Software (Bio-Rad, Hercules, CA, USA). The primers are shown in Table 2. The relative expression of all genes was calculated by the 2^−ΔΔCT^ method [33], using *S. aurata 18S* and *ef1α* as the endogenous reference.

### 2.4. Fish Sampling and Processing

Analyses of gilthead seabream fillets fed with the experimental diets were carried out at two times: at the beginning of the storage (day 0) and after 110 days of storage at −20 °C (day 110). At day 0, all the fish were weighed, gutted and filleted. The right fillets were allotted to the analyses at day 0, while the left fillets were immediately frozen at −20 °C and then stored until day 110. Irrespective of sampling time, all the fillets were subdued to the analyses of physical and chemical parameters hereafter described.

### 2.5. Analyses of Physical Parameters

The fillet pH, color, texture, and water holding capacity (WHC) were analyzed. Fillet pH and color values were measured on triplicate positions (cranial, medial, and caudal) through a pH-meter SevenGo SG2™ (Mettler-Toledo, Schwerzenbach, Switzerland) and a Minolta CR-200 Chroma Meter (Konica Minolta, Chiyoda, Japan), respectively. The color parameters were recorded as *L** (lightness), *a** (redness index) and *b** (yellowness index) values following the CIELab system [34]. Texture was assessed as the maximum shear force value obtained utilizing the Warner-Bratzler shear blade (width of 7 cm) by a Zwick Roell^®^ 109 texturometer (Zwick Roell, Ulm, Germany), equipped with a 1 kN load cell, setting the crosshead speed at 30 mm/min. Afterwards, fillets were skinned, homogenized, and utilized to determine WHC and chemical composition. The water retention capacity of the fillets at days 0 and 110 was determined by calculating, as a percentage, the loss of water after centrifugation following the method of Eide et al. [35] as modified by Hultmann and Rustad [36].

### 2.6. Analyses of Chemical Parameters

#### 2.6.1. Moisture, Total Lipids, and Fatty Acid Composition of Fillets

The moisture of minced flesh was determined in accordance with AOAC method [37]. The total lipids were obtained according to Folch et al. [38] method, then they were gravimetrically quantified. The fatty acids (FAs) were determined in the lipid extract after trans-esterification to methyl esters (FAME), using a base-catalyzed trans-esterification [39]. The FA composition was determined by gas chromatography (GC) using a Varian GC 430 gas chromatograph (Varian Inc., Palo Alto, CA, USA), equipped with a flame ionization detector (FID) and a Supelco Omegawax™ 320 m capillary column (Supelco, Bellefonte, PA, USA). The GC conditions were recovered from Secci et al. [40]. Chromatograms were recorded with the Galaxie Chromatography Data System 1.9.302.952 (Varian Inc., Palo Alto, CA, USA). FAs were identified by comparing the FAME retention time with those of the Supelco 37 component FAME mix standard (Supelco, Bellefonte, PA, USA) and quantified through calibration curves, using tricosanoic acid (C23:0) (Supelco, Bellefonte, PA, USA) as internal standard.

#### 2.6.2. Fatty Acid Composition of HBP and HBP_SFE

FAME was determined from the total lipid content [38] according to the method described by Lepage and Roy [41] and analyzed using the conditions described by Messina et al. [42], employing a Perkin Elmer (Waltham, MA, USA) Clarus 580 instrument, equipped using a silica capillary column (30 m × 0.32 mm, df 0.25 μm; Omegawax 320, Supelco, Bellefonte, PA, USA). Individual FAMEs were identified by comparison of known standards (mix of polyunsaturated fatty acids (PUFA) 1, PUFA 2 and PUFA 3 mixed oil; Supelco, Bellefonte, PA, USA). The FA composition of HBP and HBP_SFE is shown in Appendix A. Briefly, the predominant class of FAs in both samples was the PUFA, representing more than 60% of the total FAs. The α-linolenic acid (C18:3n-3, 42.86 ± 1.20% in HBP and 43.53 ± 0.48% in HBP_SFE) was the main PUFA present in both samples. SFA was the second FA class in order of abundance (about 27%) and its main component was the palmitic acid (C16:0, 24.82 ± 0.06% in HBP and 24.51 ± 0.20% in HBP_SFE). MUFA content was the lowest in both samples, accounting less than 12% of the total FAs.

#### 2.6.3. Fillet Oxidative Status

Primary and secondary oxidation products were measured as conjugated dienes (CD) and thiobarbituric acid reactive substances (TBARS), respectively, according to the spectrophotometric methods [43,44]. The results were expressed as mmol hydroperoxides (mmol Hp/100 g fillet) and malondialdehyde equivalents (mg MDA-eq./kg fillet) for CD and TBARS, respectively.

#### 2.6.4. Antioxidant Capacity

The antioxidant capacity of the gilthead seabream fillets was determined by ABTS [2,2-azino-bis acid (3-ethylbenzothiazole n-6-sulphonic acid)] and DPPH (2,2-diphenyl-1-picrylhydrazyl) assays. Samples (3 g) were extracted with 10 mL of ethanol and the ethanol extracted samples were analyzed for ABTS reducing activity assay according to Re et al. [45] and for DPPH scavenging activity according to Blois [46] method modified by Mancini et al. [47]. Results were expressed as mmol eq Trolox/kg.

### 2.7. Statistical Analysis

The data related to the gene expression were analyzed by a one-way ANOVA where the experimental treatment (diets: CTR, P5, P10, E0.5, E1) was the fixed effect. The data of fillets quality parameters were processed by a two-way ANOVA, using the PROC GLM of SAS [48] following the model:*Y**_ij_* = *μ* + *τ**_i_* + *β**_j_* + *γ**_ij_* + *ϵ**_ijk_*(1)
where *μ* is the overall mean response, *τ**_i_* is the effect due to the *i*-th Treatment (T: CTR, P5, P10, E0.5, and E1), *β**_j_* is the effect due to the *j*-th level of storage time (S, two levels: day 0, and day 110), *γ**_ij_* is the effect due to any interaction between the T and S, ϵ is the error.

The differences between the treatment means were identified using Tukey’s test at *p* < 0.05 [48].

## 3. Results

The main growth performance of the fish (Appendix A) did not show significant differences among groups, even if the diet with greater inclusion of raw pollen (P10) resulted in tendentially worsened performance.

### 3.1. Gene Expression on Seabream Liver

The expression of some markers related to immunity was analyzed in seabream liver fed the different diets for one month (Figure 1). In specimens fed E1 diet, the relative expression of *il-1β, il-6* and *il-8* genes was significantly up-regulated compared to CTR (*p* < 0.05) fish, while fish fed P5 and P10 showed a down-regulation of *il-8* gene compared to the CTR ones (*p* < 0.05).

### 3.2. Fillet Physical Parameters

The physical characteristics of the fillets (Table 3) were not significantly (*p* > 0.05) affected by the diet treatments, but they were significantly (*p* < 0.05) affected by the frozen storage. Regarding fillet weight, it was not influenced by either diet or freezing, ranging from 67.12 g (E0.5) to 74.91 g (CTR).

According to Minolta color measurements, the *L**, *a** and *b** values were not affected (*p* > 0.05) by the diets. Instead, the frozen storage affected (*p* < 0.05) the fillet color by increasing the lightness (*L**; *p* < 0.0001) and yellowness index (*b**; *p* = 0.0072) and decreasing the redness one (*a**; *p* < 0.0001).

At day 0, the fillet pH was slightly lower (*p* = 0.0003) than the one recorded at the end of the storage period.

Concerning the WHC, it was pronouncedly higher (*p* < 0.0001) in day 0 fillets compared with the fillets analyzed at day 110.

Despite the shear force not resulting in a statistical difference (*p* = 0.36) among the experimental groups, the *p*-value (0.064) revealed that the shear force tended to increase, after a period of 110 days of frozen storage.

### 3.3. Fillet Chemical Parameters

#### 3.3.1. Chemical and Fatty Acid Composition

Table 4 shows the results of the chemical composition and FA profile of the gilthead seabream fillets. The moisture of the fillets did not differ between the dietary groups (*p* = 0.788) whilst it was deeply reduced (*p* < 0.0001) by the storage. Even the total lipid content was not affected (*p* = 0.841) by the HBP inclusion in the diets but was reduced (*p* = 0.0026) after the storage period.

Oleic (C18:1n-9), linoleic (C18:2n-6, LA) and palmitic (C16:0) acids were the major FAs in all the groups, representing on average 32.31%, 22.04% and 12.41% in the dietary treatments, respectively. Overall, monounsaturated fatty acids (MUFAs) were abundantly present in all the dietary treatments (from 42.13 to 42.66%) and were not affected by the diets (*p* = 0.396). The second most abundant FA group was the n-6 PUFA, ranging from 23.75 to 24.12%, which was also not affected by the diets (*p* = 0.722). While saturated fatty acids (SFA) and n-3 PUFAs were not affected by the diets (*p* = 0.399), they were after the storage period of 110 days that increased the SFA content and decreased the n-3 PUFA (*p* = 0.002) one. The α-linolenic (C18:3n-3, ALA) and the eicosenoic (C20:1n-9) acids were affected both by the diets and by the frozen storage. At the beginning of the storage, the E1 fillets presented the highest value of ALA, while the highest values of C20:1n-9 were found in both the P5 and E1 fillets. The incidences of DHA (C22:6n-3) and EPA (C20:5n-3) were not affected (*p* > 0.05) by the diets nor by the storage period.

#### 3.3.2. Fillet Oxidative Status

The results of fillet lipid oxidation (Table 5) revealed that the diets and the storage did not significantly affect (*p* = 0.628 and *p* = 0.322, respectively) the CD content of the fillets. Instead, the degree of lipid oxidation, in terms of TBARS, was not affected by the dietary treatments (*p* = 0.409) whilst it was deeply affected (*p* < 0.0001) by the storage, with values increased after 110 days of storage.

A significant interaction between the dietary treatment and the storage emerged for ABTS and DPPH (Figure 2 and Figure 3, respectively). The E0.5 fillets at day 110 presented the lowest values of ABTS while the E1 fillets presented the highest and the lowest values of DPPH at day 0 and 110, respectively. The CTR fillets were affected in terms of DPPH value (Figure 3). Of note, in the five groups of fillets coming from differently fed gilthead seabream, the storage period at low temperatures (−20 °C) resulted in a decrease in the values of both parameters (ABTS and DPPH) compared to the values obtained for day 0.

## 4. Discussion

The quantity of pollen ingested by the fish in this trial calculated from the feed ingestion (see Appendix A) was 0.9 g/kg of average body weight per day and 0.43 g/kg of average body weight per day for P10 and P5 diet, respectively. Growth performance of the fish did not show significant differences among groups, even if the diet with greater inclusion of raw HBP resulted in tendentially worsened performance. This seemed to confirm what was already observed by Panettieri et al. [15] where raw HBP-containing diets were offered to meagre juveniles impairing growth performance indexes and nutrient digestibility as the level of HBP in the diet increased. The authors also found alterations of medium intestine histology, immunohistochemistry, blood biochemical parameters and hepatic biomolecular markers, following the same trend. The authors attributed these negative effects to the ultrastructure of the bee pollen grains walls that makes the bioactive substances unavailable and can irritate the intestine of carnivorous fish [13]. These unfavorable events can be overcome by extraction procedures that make the bioactive components available to the fish while at the same time discarding the outermost layer composed of neutral lipids, hydrocarbons, terpenoids and carotenoid pigments and reinforced inside by exine, a matrix of complex carbohydrate, sporopollenin that greatly resists to monogastric digestion [8].

Bioactive compounds by HBE, such as essential amino acids, fatty acids, vitamins and flavonoids, improve immune response [49]. It has been shown that peony flower extract could increase physiological, anti-aging and antioxidant activity by improving immune response in fish [50,51]. Although immunostimulatory properties have been proposed for bee pollen in several species, the effect of dietary supplementation with HBP_SFE on immune response and fillet quality has not been previously described. In a recent study, Ren et al. [23] observed that peony pollen diet supplement significantly promoted the growth performance, improved intestinal health and digestive enzymes activity of common carp (*Cyprinus carpio*). Furthermore, they suggested that the use of peony pollen could improve the fish health status by increasing highly unsaturated fatty acids (HUFA) content, improving intestinal health, elevating immune responses and antioxidant capacity of common carp organs [23]. In the present study, the inclusion of bee pollen in the *S. aurata* diet affected the levels of some markers related to the immunity of fish and some quality parameters of fillets. Among immune-related gene expression, *il-1β*, *il-6* and *il-8* were selected because they are related to inflammatory processes. In teleost, *il-1β* is probably the most widely studied of the known cytokines, and has been characterized in a number of fish species [52]. The *il-1β* has been found to be regulated in response to various *stimuli* and the biological activity of recombinant *il-1β* (*ril-1β*) has been studied in several fish species indicating that fish *il-1β* is involved in the regulation of immune relevant genes, lymphocyte activation, migration of leucocytes, phagocytosis and bactericidal activities [53,54,55]. The important pleiotropic cytokine *il-6* plays an essential role in mediating innate and adaptive immune responses [56]. The first *il-6* sequence identified in teleost was reported in fugu (*Takifugu rubripes*) [57] and further characterized in other fish species such as flounder (*Paralichthys olivaceus*) [58] and gilthead seabream (*Sparus aurata*) [59]. The *il-8* is a well-recognized pro-inflammatory chemokine that attracts neutrophils and other leukocytes, promoting their recruitment to sites of inflammation [60]. In fish fed both diets supplemented with P5 or P10, results showed some inhibitory effects in the immune system, such as the downregulation in the *il-8* expression from the liver. The decrement of *il-8* expression could be understood as a decrease in the immune response that could entail susceptibilities to pathogens. In fact, *il-8* downregulation observed in the present study may partially be attributed to the inhibiting effects of pollen on the phosphorylation of inhibitor kappa B protein and mitogen-activated protein kinases [61,62]. Our previous paper attested a decrease of serum peroxidase and lysozyme activities, suggesting that the inclusion of 5% or 10% of bee pollen in the diet of seabream may impair the defense capacity [8].

In addition, a diet supplemented with the highest concentration of bee pollen extract (E1) induced the up-regulation in all the studied genes in the liver, i.e., *il-1β*, *il-6*, and *il-8*. These results indicate that the pollen extract activated the seabream immune system, according to previous studies in which the immunostimulant efficacy of plants, certain parts of the same, or even their extracts used in feeding tests on fish has been demonstrated [63,64,65,66,67]. Messina et al. [8] demonstrated that diets supplemented with SFE bee pollen extract had a stimulating effect on fish serum immunity compared with the inclusion of raw pollen in agreement with our results.

As consumers are increasingly interested in food products associated with animal welfare and concerned with general quality, many studies are working with a multidisciplinary approach to evaluate the effects of new additives that fit these new market demands. For fish fillets or meat in general, one of the most perceivable characteristics is the color, hence, an essential parameter to be analyzed [3,68,69].

The results of *L** (lightness), *a** (redness index) and *b** (yellowness index) of fillets were similar to those presented by Ünal Şengör et al. [70], that evaluated color of commercial gilthead seabream during cold storage. Whereas no information available about the possible effects of HBP on fish fillet quality and its relationship with the immune response, in a study using a natural compound (*Flammulina velutipes* extract) in catfish diets, it was observed that the experimental diets prevented lipid oxidation and improved color stability during frozen storage [69].

The pH and WHC values of the fish fillets at day 0 were unaffected by the dietary intervention, according to Álvarez et al. [71]. On the contrary, storage time showed a significant effect on those parameters. The pH increase after the storage period is associated with the accumulation of components from the nitrogen compounds degradation [72,73]. Concerning the WHC, the decreased value can be attributable both to the muscle mechanical damage by the ice crystal formation during freezing and to the protein denaturation and lipid oxidation as well [40,74,75,76,77], in line with other studies [36,76,77,78]. Texture instead, expressed as shear force, was not affected nor by treatment or frozen storage in gilthead seabream fillets and the results obtained were similar to Cai et al. [79] results.

Moisture and total lipids of the fish fillets were unaffected by the dietary intervention; however, storage time significantly decreased water content, as herein found. This effect on water content can be explained because of the damages to the cells caused by ice crystals formed during freezing. Thus, the free water is readily separated from the matrix after thawing, resulting in a great loss of water [80]. As suggested by Sathivel [81], the effect of moisture loss during frozen storage can be attenuated if some type of protection is used. For instance, different coatings, such as chitosan, egg albumin or soy protein concentrate effectively preserved fish fillets from water loss [81], while the dietary intervention with HBP did not represent a valid approach.

The total lipid content, unaffected by the diet, was significantly decreased (*p* < 0.05) after 110 days of storage at −20 °C, as previously observed in other fish species [69,77,82,83,84,85].

Regarding the fatty acid composition of HBP and HBP_SFE, among the predominant fatty acids in pollen extract, PUFA was the most abundant. Xu et al. [14] highlighted that extracted bee pollen oil has high nutritional quality and is, therefore, an excellent functional oil. It is known that PUFAs are critical components of cell and organelle membranes (mainly as *sn*-2 phospholipids), as suggested by the close link between organelle evolution in eukaryotes and the emergence of polyunsaturated fatty acids [86]. In addition, PUFAs also regulate the expression of certain genes, including those coding for fatty-acid synthase, nitric-oxide synthase, sodium-channel proteins and cholesterol-7-a-hydroxylase and thereby they affect processes including fatty-acid biosynthesis, cancer induction and cholesterol regulation [87]. In this way, they have an impact on cellular biochemical activities, transport processes and cell-stimulus responses, are implicated in physiological processes including lipid metabolism and targeting, immune responses and cold adaptation, and are involved in pathological conditions such as carcinogenesis and cardiovascular disease [86,87,88]. Nevertheless, the quantity of HPB and SFE added to the diet was not able to determine the significant modification of the fillet’s fatty acid profile. Nor EPA or DHA, two fatty acids having a very important structural and functional roles in fish metabolism [89], were significantly modified by the treatment. However, the content of C18:3n-3, the fatty acid precursor of EPA and DHA, found higher in the E1 group compared to the other diets, could deserve further investigations to better understand the interaction between HBP and the expression of genes involved in lipid metabolism. Indeed, peony pollen administrated for 6 weeks to common carp might regulate fish lipid metabolism and fatty acids profile [23]. Concerning lipid stability, the frozen storage slightly increased the relative abundance of myristic (C14:0), palmitic (C16:0) and palmitoleic (C16:1n-7) acids. In addition, the frozen storage also affected the levels of α-linolenic (C18:3n-3), docosapentaenoic (C22:5n-3, DPA), linoleic (C18:2n-6, LA) and 9-eicosenoic (C20:1n-9) acids by slightly decreasing them. These moderated changes agreed with data from lipid peroxidation, especially those of TBARS content. This trend was also confirmed by DPPH values for all the groups, with a minor effect of the diets. Otherwise, despite not differing significantly, the fillets of the groups that received diets with high concentrations of HBP (P10 and E1) apparently were less affected by oxidative stress, compared to the control group, as far as concerns the secondary oxidation products. This trend was also associated with ABTS values, as P10, E1 and P5 showed lower decreasing values than CTR and E0.5. Herein, the present result did not support previous evidence regarding the beneficial effects on the overall antioxidant capacity and malondialdehyde content of the intestine of common carp fed with 30 g/kg of peony pollen [23]. Indeed, the antioxidant capacity expressed in E1, P10 and P5 was not sufficient to overdue the lipid oxidation caused by the storage.

## 5. Conclusions

Considering all the data obtained in this trial and the growing attention towards animal welfare, we can conclude that the pollen supercritical fluid extract added to the diet for seabream at 1% seemed a promising option due to the positive effect on hepatic expression of inflammatory genes. The research carried out demonstrated that the intrinsic bioactivity of the honeybee pollen was properly extracted by the supercritical fluid technique and exerted its bioactivity in vivo, thanks to some beneficial effects on fish gilthead seabream health. Contrarily, no specific protection of HBP against lipid peroxidation was shown during 110 days of frozen storage. For this reason, further studies are highly suggested to verify if higher pollen inclusion levels, a mix of HBP and SFE, or longer feeding trials than the present one could positively affect the fatty acid profile and mitigation of fillet oxidative damages. The present study suggests the need for a multidisciplinary approach in the research aimed at discovering natural compounds able to modulate fish welfare and fillet quality.

## Figures and Tables

**Figure 1 animals-12-00675-f001:**
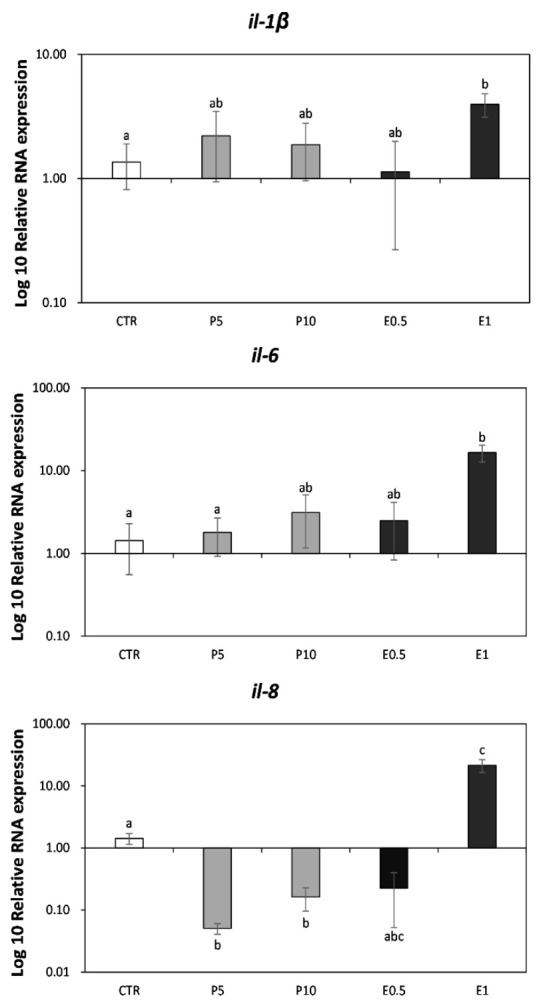
Relative expression of genes related to the immune system analyzed on liver of gilthead seabream. CTR: control diet (without HBP and HBP_SFE); P5 and P10: diets including 5% or 10% of honeybee pollen (HBP), respectively; E0.5 and E1: diets including 0.5% or 1% supercritical fluid extract (SFE) from HBP (HBP_SFE), respectively. Values are the mean ± SEM (n = 9). Statistical differences (*p* < 0.05) between groups are indicated by the different letters (a, b, c).

**Figure 2 animals-12-00675-f002:**
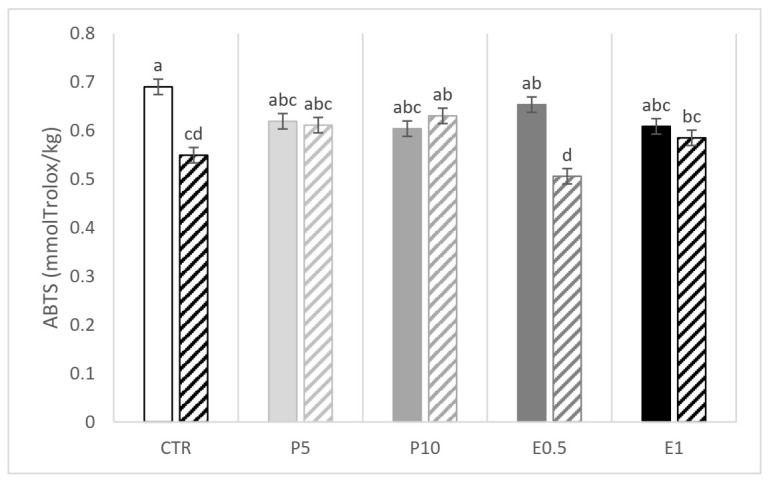
ABTS values (mmol eq. Trolox/kg fillet) of fillets from fish fed the CTR: control diet without HBP and HBP_SFE (white bars); P5 (light grey bars) and P10 (grey bars): diets including 5% or 10% of honeybee pollen (HBP), respectively; E0.5 (dark grey bars) and E1 (dark bars): diets including 0.5% or 1% of supercritical fluid extract (SFE) of HBP (HBP_SFE), respectively. Full colored bars are for samples analyzed at day 0, while crossed bars are for samples analyzed at day 110. The letters a, b, c, d indicate statistically different means according to the significant T × S interaction (*p* < 0.05).

**Figure 3 animals-12-00675-f003:**
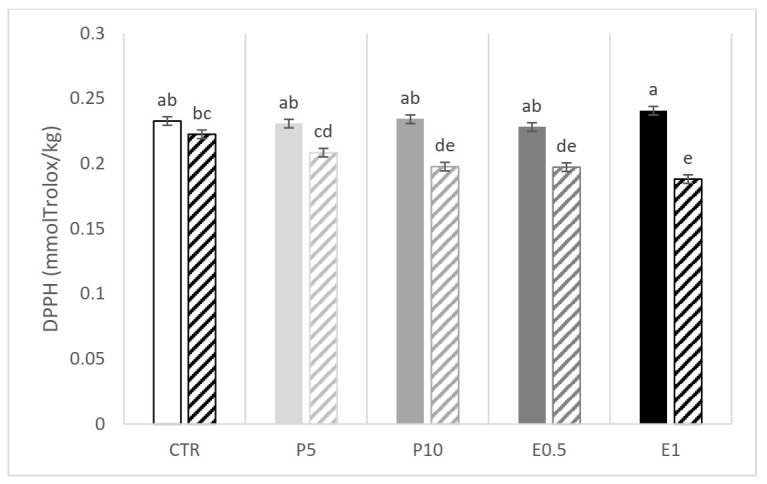
DPPH value of fillets from fish fed the CTR: control diet without HBP and HBP_SFE (white bars); P5 (light grey bars) and P10 (grey bars): diets including 5% or 10% of honeybee pollen (HBP), respectively; E0.5 (dark grey bars) and E1 (dark bars): diets including 0.5% or 1% of supercritical fluid extract (SFE) of HBP (HBP_SFE), respectively. Full colored bars are for samples analyzed at day 0, while crossed bars are for samples analyzed at day 110. The letters a, b, c, d, e indicate statistically different means according to the significant T × S interaction (*p* < 0.05).

**Table 1 animals-12-00675-t001:** Chemical composition (% as feed) of the experimental diets provided to gilthead seabream.

	CTR	P5	P10	E0.5	E1
Dry matter	88.94	88.19	87.63	88.43	88.62
Ash	5.50	4.99	4.35	4.71	4.51
Crude protein	39.78	39.92	38.26	38.91	39.46
Ether extract	17.85	17.25	17.15	17.75	17.15
Crude fiber	7.17	7.54	8.63	7.37	7.62

CTR: control diet (without HBP and HBP_SFE); P5 and P10: diets including 5% or 10% of honeybee pollen (HBP), respectively; E0.5 and E1: diets including 0.5% or 1% supercritical fluid extract (SFE) of HBP (HBP_SFE), respectively.

**Table 2 animals-12-00675-t002:** Gilthead seabream primer sequences utilized for the real-time PCR.

Gene	Accession Number	F/R Primer Sequence (5′–3′)
*il1-β*	AJ277166	F-GGGCTGAACAACAGCACTCTC
		R-TTAACACTCTCCACCCTCCA
*il-6*	AM749958	F-AGGCAGGAGTTTGAAGCTGA
		R-ATGCTGAAGTTGGTGGAAGG
*il-8*	AM765841	F-GCCACTCTGAAGAGGACAGG
		R-TTTGGTTGTCTTTGGTCGAA
*ef1α*	AF184170	F-CTTCAACGCTCAGGTCATCAT
		R-GCACAGCGAAACGACCAAGGGGA
*18S*	AM490061	F-CTTCAACGCTCAGGTCATCAT
		R-AGTTGGCACCGTTTATGGTC

**Table 3 animals-12-00675-t003:** Physical characteristics of fillets from gilthead seabream fed the different dietary treatments analyzed before (day 0) and after 110 days of frozen storage.

	Treatment, T					Storage, S		*p*-Value		RMSE
	CTR	P5	P10	E0.5	E1	0	110	T	S	
Weight, g	74.91	72.45	67.42	67.12	71.56	71.70	69.68	0.835	0.688	19.347
*L**	53.11	54.10	53.87	53.88	54.47	48.65	59.13	0.725	<0.0001	2.391
*a**	−1.75	−1.73	−1.51	−1.89	−1.84	−0.48	−3.01	0.790	<0.0001	0.764
*b**	0.20	0.78	−0.06	−0.29	−0.66	−0.53	0.52	0.172	0.0072	1.450
pH	6.06	6.06	6.14	6.11	6.06	6.04	6.14	0.163	0.0003	0.098
Shear force, N	43.12	41.33	49.18	40.57	44.12	40.94	46.39	0.360	0.064	11.127
WHC, %	73.31	77.93	76.17	78.43	78.48	86.87	66.85	0.608	<0.0001	9.165

CTR: control diet (without HBP and HBP_SFE); P5 and P10: diets including 5% or 10% of honeybee pollen (HBP), respectively; E0.5 and E1: diets including 0.5% or 1% of supercritical fluid extract (SFE) of HBP (HBP_SFE), respectively. WHC: water holding capacity. ns: not significant (*p* > 0.05); the interaction T × S is not reported because not significant (*p* > 0.05). RMSE: Root Mean Square Error.

**Table 4 animals-12-00675-t004:** Moisture, total lipids (g/100 g of fillet) and fatty acid profile (g FA/100 g FAME) of fillets from gilthead seabream fed the different dietary treatments, analyzed before (day 0) and after 110 days of frozen storage.

	Treatment, T					Storage, S		*p*-Value		RMSE
	CTR	P5	P10	E0.5	E1	0	110	T	S	
Moisture	61.35	60.17	63.05	62.61	61.27	67.58	55.81	0.788	<0.0001	6.100
Total lipids	9.31	9.79	8.75	8.81	9.16	10.11	8.22	0.841	0.003	2.26
Fatty acids										
C14:0	2.18	2.19	2.18	2.21	2.16	2.10	2.27	0.878	<0.0001	0.11
C16:0	12.43	12.48	12.25	12.52	12.35	12.07	12.74	0.461	<0.0001	0.41
C16:1n-7	3.56	3.53	3.53	3.60	3.49	3.46	3.62	0.587	0.0003	0.16
C18:0	3.10	3.10	3.15	3.03	3.15	3.09	3.12	0.245	0.422	0.15
C18:1n-9	32.41	32.14	32.34	32.56	32.10	32.35	32.28	0.144	0.604	0.50
C18:1n-7	2.69	2.69	2.66	2.71	2.68	2.69	2.68	0.212	0.535	0.06
C18:2n-6	22.01	22.02	22.17	21.91	22.07	22.10	21.98	0.872	0.007	0.01
C18:3n-3	3.86 ^ab^	3.90 ^ab^	3.84 ^b^	3.85 ^b^	3.99 ^a^	3.93	3.84	0.026	0.008	0.12
C20:1n-9	1.55 ^ab^	1.62 ^a^	1.47 ^b^	1.53 ^ab^	1.61 ^a^	1.59	1.52	0.001	0.004	0.08
C20:5n-3	2.39	2.35	2.43	2.38	2.47	2.44	2.36	0.185	0.183	0.13
C22:5n-3	1.23	1.24	1.27	1.20	1.23	1.29	1.18	0.373	<0.0001	0.08
C22:6n-3	5.58	5.42	5.70	5.43	5.59	5.64	5.45	0.574	0.104	0.46
Σ SFA	18.56	18.64	18.46	18.59	18.53	18.13	18.98	0.869	<0.0001	0.47
Σ MUFA	42.42	42.36	42.13	42.66	42.21	42.37	42.34	0.396	0.891	0.67
Σ n-6PUFA	23.86	23.93	24.12	23.81	23.86	24.08	23.75	0.722	0.405	0.58
Σ n-3PUFA	14.41	14.29	14.54	14.20	14.64	14.68	14.16	0.399	0.002	0.61

CTR: control diet (without HBP and HBP_SFE); P5 and P10: diets including 5% or 10% of honeybee pollen (HBP), respectively; E0.5 and E1: diets including 0.5% or 1% of supercritical fluid extract (SFE) of HBP (HBP_SFE), respectively. SFA: saturated fatty acids; MUFA: monounsaturated fatty acids; PUFA: polyunsaturated fatty acids. ns: not significant (*p* > 0.05); a, b: means with different superscript letters are significantly different (*p* < 0.05) among diets; the interaction T × S is not reported because not significant (*p* > 0.05). RMSE: Root Mean Square Error. The following FAs, found below 1% of the total FAME, were utilized for calculating the Ʃ classes of FAs but they are not listed: C12:0, C13:0, C14:1n-5, C15:0, C16:1n-9, C16:2n-4, C16:3n-4, C17:0, C17:1, C18:2n-4, C18:3n-4, C18:4n-1, C18:4n-3, C20:0, C20:1n-11, C20:1n-7, C20:2n-6, C20:3n-3, C20:3n-6, C20:4n-3, C21:5n-3, C22:0, C22:1n-7, C22:1n-9, C22:1n-11, C22:2n-6, C22:4n-6, C22:5n-6, C24:0.

**Table 5 animals-12-00675-t005:** Conjugated dienes (CD, mmol Hp/100 g fillet), and TBARS content (mg MDA-eq./kg fillet) determined on fillets from gilthead seabream fed the different dietary treatments analyzed before (day 0) and after 110 days of frozen storage.

	Treatment, T					Storage, S		*p*-Value		RMSE
	CTR	P5	P10	E0.5	E1	0	110	T	S	
CD	0.262	0.274	0.263	0.237	0.230	0.243	0.263	0.628	0.322	0.079
TBARS	0.863	0.979	0.762	0.913	0.858	0.627	1.123	0.409	<0.0001	0.271

CTR: control diet (without HBP and HBP_SFE); P5 and P10: diets including 5% or 10% of honeybee pollen (HBP), respectively; E0.5 and E1: diets including 0.5% or 1% of supercritical fluid extract (SFE) of HBP (HBP_SFE), respectively. ns: not significant (*p* > 0.05); the interaction T × S is not reported because not significant (*p* > 0.05). RMSE: Root Mean Square Error.

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
