# Peer review of "Effects of Dietary Supplementation with Honeybee Pollen and Its Supercritical Fluid Extract on Immune Response and Fillet’s Quality of Farmed Gilthead Seabream (Sparus aurata)"

_animals, 2022, doi:10.3390/ani12060675_

Round 1

Reviewer 1 Report

This manuscript reported the effects of dietary supplementation with honeybee pollen and its supercritical fluid extract on immune response and fillet's  quality of farmed gilthead seabream (Sparus aurata).The topic about this work covered could meet the scope of this journal. As far as I concerned, this study provides some new information and data for the application of honeybee pollen in gilthead seabream breeding. Although the topic of the study is valuable, the manuscript presents several problems. Before the formal acceptance, the author is expected to be able to seriously and extensively revise.

Abstract:

1.The abstract is not well written and the main content of the article is not clearly stated. Rewrite the abstract, the results are too few, follow the format of the research meaning, method, result, conclusion, focus on the results.

Introduction

1.Line 56, delete the “both”.

2.Line 59, add“to improve health production”after “tool”.

3.Line 60-61, This sentence is not clearly expressed. I suggest that it should be rewritten.

4.Line 82-86, It is recommended to enumerate the results of studies on feeding effects such as growth performance, immunity, body colour and meat quality.

5.Line 87-97, This paragraph is not written in a clear way. I suggest you rewrite it.

Materials and Methods

1.Line 108-109, How was the amount of addition of HBP and HBP-SFE determined? Is there a reference? What are the main active ingredients of the HBP-SFE?

2.Merge tables 1 and 2.

3.Line 142, delete “for 142 seven days a week”.

4.Line 226, “Re et al. to [48]” It doesn’t match the references.

Results

1.In table 4. How to get the data of fillet?

  1. The “3.3.2. Fatty acid composition of HBP and HBP_SFE” Recommend moving to “Materials and Methods”.

Discussion and conclusion

This part of the written tedious, incoherent, I suggest to rewrite.

Author Response

Dear Editor,

thanks to the anonymous reviewer for their comments and suggestions. Please consider the following answers aimed to solve the criticisms highlighted. Several changes have been made in the manuscript, both in the Introduction and the Discussion sections. All the reference list has been updated. We hope that now the manuscript has raised the high-quality standard required by the Journal.

Reviewer 1

This manuscript reported the effects of dietary supplementation with honeybee pollen and its supercritical fluid extract on immune response and fillet's quality of farmed gilthead seabream (Sparus aurata). The topic about this work covered could meet the scope of this journal. As far as I concerned, this study provides some new information and data for the application of honeybee pollen in gilthead seabream breeding. Although the topic of the study is valuable, the manuscript presents several problems. Before the formal acceptance, the author is expected to be able to seriously and extensively revise.

Abstract:

R#1 The abstract is not well written and the main content of the article is not clearly stated. Rewrite the abstract, the results are too few, follow the format of the research meaning, method, result, conclusion, focus on the results.

Authors: Thank you for your comment. The abstract has been deeply revised.

Introduction

R#1 Line 56, delete the “both”.

Authors: The correction has been done.

R#1 Line 59, add “to improve health production” after “tool”.

Authors: The correction has been done.

R#1 Line 60-61, This sentence is not clearly expressed. I suggest that it should be rewritten.

Authors: The correction has been done, and the overall introduction has been improved as suggested by the other reviewers.

R#1 Line 82-86, It is recommended to enumerate the results of studies on feeding effects such as growth performance, immunity, body colour and meat quality.

Authors: The correction has been done.

R#1 Line 87-97, This paragraph is not written in a clear way. I suggest you rewrite it.

Authors: Thank you for your suggestions. The paragraph has been modified.

Materials and Methods

R#1 Line 108-109, How was the amount of addition of HBP and HBP-SFE determined? Is there a reference? What are the main active ingredients of the HBP-SFE?

Authors: The results presented in this article are part of a wider series of studies on the use of honey bee pollen in fish species whose results have been already published (see Panettieri et al., 2020, cit.). In particular, the preliminary results of this trial have been reported in the following publication: Messina CM, Panettieri V, Arena R, Renda G, Espinosa Ruiz C, Morghese M, Piccolo G, Santulli A and Bovera F (2020) The Inclusion of a Supercritical Fluid Extract, Obtained From Honey Bee Pollen, in the Diet of Gilthead Sea Bream (Sparus aurata) Improves Fish Immune Response by Enhancing Anti-oxidant, and Anti-bacterial Activities. Front. Vet. Sci. 7:95. doi: 10.3389/fvets.2020.00095. The inclusion levels have been in fact stated on the basis of the cited article. The text was modified accordingly.

The main active substances of HBP-SFE are polyphenols as reported in Messina et al. (2020) and already indicated in the text.

R#1 Merge tables 1 and 2.

Authors: Thank you for your suggestion. In our opinion it could be quite unclear a merged table hence we preferer to list the ingredient in the appropriate section of M&M and to leave only Table 2. Hope the R#1 agree with us.

R#1 Line 142, delete “for 142 seven days a week”.

Authors: The correction has been done.

R#1 Line 226, “Re et al. to [48]” It doesn’t match the references.

Authors: Thank you. All the reference list has been updated due to the new insertions. Re et al. has been inserted [45].

Results

R#1 In table 4. How to get the data of fillet?

Authors: We are sorry but we have not clear what does it R#1 refer to. We have modified the item “fillet” that could be unclear with “weight”. We hope to have answered to R#1.

R#1 The “3.3.2. Fatty acid composition of HBP and HBP_SFE” Recommend moving to “Materials and Methods”.

Authors: The correction has been done and the Table reporting HPB fatty acids has been moved in the Supplementary Material (Table S1).

Discussion and conclusion

R#1 This part of the written tedious, incoherent, I suggest to rewrite.

Authors: We have deeply modified the discussion according to reviewers’ opinion and suggestions. We hope that now it appears more interesting and coherent than the original version.

Reviewer 2 Report

This is an interesting study about the. Effects of dietary supplementation with honeybee pollen and its supercritical fluid extract on immune response and fillet's quality of farmed gilthead seabream (Sparus aurata). I could see a complete analysis about the immune response and fillets’quality in farmed S aurata, however, the authors should point out some extra issues and make some corrections in order to publish this work in this prestigious journal.

Abstract

Line 33. In the abstract it is important to mention the methodology development of the culture and culture conditions, likewise the different tissue analyzed. In addition, the methodology of the expression.

Introduction

It is necessary that the MS emphasize the importance of this study on the importance of the supplementation of different source of protein in specie-specific, in special the contribution to the immune response and function related with the physiology of this specie. In addition, to justify the importance of evaluating the expression of genes. Is important that the introduction rewriting to include the aims of the work properly.

Please mention some previous studies where they relate the immune response, fillet´s quality with the gene expression.

In the MS the authors only mention SBP but not SFE. What is the importance of SFE in the diet?

What is the effect of SBP and SFE on fillet quality, what is their relationship?

Materials and Methods

Line 107. Specify the level protein and energy in diets (Isoenergetic and isoproteic)

Line 107. The control diet without HBP and SFE?

Line. Describe the process of preparing the experimental diets

Line 125. The CTR was prepared in the laboratory or commercial? (Table 1)

Line 125. Micronutrients? (Table 1)

Line 130. Table 2. Isoenergetic? Put the energy level (J, KJ, cal, etc). Ether extract is Isolipidc.

Line. 143. 30 days experimental. Why? Justify.

Line 144. photoperiod?

line 144. Time of sampled? Before or after the first feeding?

Line. If you have weight (mg) data, please put them in the methodology

line 146. Please specify dilution factor tissue:PUREzol Reagent

Line 240. Edit the formula

Line 221. Aminoacid?  I considered that que aminoacid profile is very important in the present MS.

Line . Why were other treatments with HBP and SFE (mix) not performed?

Results

1.Growth (weight and length)? Figure ??

  1. I consider that growth performance in this study are important, because we look for new sources of protein in diets for fish
  2. Please indicate significant differences with values (P = ??)
  3. Homologate the size of the images so that it has better resolution.
  4. Analysis of hematological and biochemical blood are important in study on the effect of supplementation of different source of protein in fish on immune response. Justify.

Discussion

  1. The discussion has been written in a very simplistic way. However, you’ll need to discuss comparing with similar fish species and similar topics. You’ll need to reinforce aspect about the effect of dietary supplementation with HBP and SFE on immune response and fillet´s quality with the relationship the gene expression in specific tissues.

I believe that the manuscript needs major revision by the authors before being published in this prestigious journal

Author Response

Reviewer 2

This is an interesting study about the Effects of dietary supplementation with honeybee pollen and its supercritical fluid extract on immune response and fillet's quality of farmed gilthead seabream (Sparus aurata). I could see a complete analysis about the immune response and fillets’ quality in farmed S aurata, however, the authors should point out some extra issues and make some corrections in order to publish this work in this prestigious journal.

Abstract

R#2 Line 33. In the abstract it is important to mention the methodology development of the culture and culture conditions, likewise the different tissue analyzed. In addition, the methodology of the expression.

Authors: The abstract has been corrected following the suggestions of all the referees. For example, the methodology of the gene expression and the analyzed tissues; however, we could not insert all the details since the Journal ask for 200 words, we hope that the corrections done could satisfy the Reviewer.

Introduction

R#2 It is necessary that the MS emphasize the importance of this study on the importance of the supplementation of different source of protein in specie-specific, in special the contribution to the immune response and function related with the physiology of this specie. In addition, to justify the importance of evaluating the expression of genes. Is important that the introduction rewriting to include the aims of the work properly.

Authors: thank you for suggestion, we modified the text in the Introduction section and also in the other sections.

R#2 Please mention some previous studies where they relate the immune response, fillet´s quality with the gene expression.

Authors: thank you for suggestion, the introduction has been modified in order to highlight the novelty and the aim of the study.

R#2 What is the effect of SBP and SFE on fillet quality, what is their relationship?

Authors: We agree with R#2, no a clear relationship between pollen and quality was present in the introduction. As a consequence, the introduction has been modified.

Materials and Methods

R#2 Line 107. Specify the level protein and energy in diets (Isoenergetic and isoproteic)

Authors: done

R#2 Line 107. The control diet without HBP and SFE?

Authors. Thank you for that comment. Yes, the control diet was without HBP and SFE. We have now specified that in the text.

R#2 Line. Describe the process of preparing the experimental diets

Authors: The process is described in detail (see lines 172-212)

R#2 Line 125. The CTR was prepared in the laboratory or commercial? (Table 1)

Authors: All the diets were prepared in the laboratory. It has been now better specified in the text revised.

R#2 Line 125. Micronutrients? (Table 1)

Authors: The micronutrients analyses of the diets have not been performed. Although we know the micronutrients composition of the mineral and vitamin mixes that are as follows:

Mineral mix supplying g/kg diet, CaHPO4+2H2O, 1.50, KH2PO4, 5.00, NaCl, 0.04, MgO, 2.50, FeCO3, 0.70, KI, 0.04, ZnO, 0.11, MnO, 0.10, CuSO4, 0.01, Na Selenite, 0.0004.

Vitamin mix supplying mg or IU/kg diet: vit. A, as retinyl palmitate 5000 IU; vit. D3, 2400 IU; α-tocopheryl acetate, 350; menadione, 50; thiamin HCl, 40; riboflavin, 50; pyridoxine HCl, 40; Ca-pantothenate 50; vit. B12, 0.01; niacin, 300; biotin, 3.0; folic acid, 5.0; choline 3750, myo-inositol, 500; vit. C as ascorbate Mg-phosphate, 200.

HBP trace elements content was reported in Panettieri et al. (2020).

R#2 Line 130. Table 2. Isoenergetic? Put the energy level (J, KJ, cal, etc.). Ether extract is Isolipidc.

Authors: Corrected in Material and Methods

R#2 Line. 143. 30 days experimental. Why? Justify.

Authors: As mentioned above, this trial is part of a wider series of studies aimed to test the possible effects of honeybee pollen on several aspects of fish growth, immune response, fish quality etc. In particular, in a first trial, we focused on how the use of different inclusion levels of raw HBP in the diet can influence growth, nutrients digestibility, intestine histology, immunohistochemistry, hepatic biomolecular markers and blood biochemical analyses in a carnivorous fish species (Panettieri et al. 2020, cit.). In this trial instead we preferred to focus our research on immune response and organoleptic quality of the fillets rather than on growth performance. This is why we started the trial with specimens with a higher average initial body weight and we brought them to commercial size in 30 days.

R#2 Line 144. photoperiod?

Authors: Added.

R#2 line 144. Time of sampled? Before or after the first feeding?

Authors: After a 24-hour fast. Added to the text.

R#2 Line. If you have weight (mg) data, please put them in the methodology

Authors: The line is missing, however we supposed that R#2 referred to the amount of liver biopsies, hence we have added this information (around 300 mg).

R#2 line 146. Please specify dilution factor tissue:PUREzol Reagent

Authors: Thanks for your suggestion; we specified the dilution factor tissue in the text.

R#2 Line 240. Edit the formula

Authors: The formula has been added.

R#2 Line 221. Aminoacid? I considered that que aminoacid profile is very important in the present MS.

Authors: Please, see the next point.

R#2 Line . Why were other treatments with HBP and SFE (mix) not performed?

Authors: We agree with R#2, it would be interesting to perform a new trial testing different concentration and mixing HPB and SFE, but it could not be possible this time. Thanks for your suggestion.

Results

R#2. Growth (weight and length)? Figure ??

Authors: Please, see the next point.

R#2 I consider that growth performance in this study are important, because we look for new sources of protein in diets for fish

Authors: The aim of this study was to investigate the effects of HBP and its extract on some parameters related to the immune response and quality, in farmed gilthead seabream (Sparus aurata). In our opinion, in fact, for a matter of quantities that can be produced worldwide, pollen will never represent a valid protein source alternative to fishmeal in aquafeeds. And therefore, HBP was here tested as a source of bioactive compounds that can have significant beneficial effects on fish immunity responses and quality. Not surprisingly, growth performance has been left in the background. However, a table was added as supplementary material showing the main growth performance of fish that did not show significant differences among groups, even if the diet with a greater inclusion of raw pollen resulted in tendentially worsened performance, confirming what already observed in Panettieri et al. (2020). Aminoacid profile analysis was not performed in this trial.

R#2 Please indicate significant differences with values (P = ??)

Authors: The correction has been done and, consistently, the Tables have been adjusted.

R#2 Homologate the size of the images so that it has better resolution.

Authors: The correction has been done

R#2 Analysis of hematological and biochemical blood are important in study on the effect of supplementation of different source of protein in fish on immune response. Justify.

Authors: Dear Referee, we agree with your comment, in fact, as we have in plan to do again a trial with this compound to evaluate other performance, we will include these analyses in the next experiment, as unfortunately it was not possible to do in the present study.

Discussion

R#2 The discussion has been written in a very simplistic way. However, you’ll need to discuss comparing with similar fish species and similar topics. You’ll need to reinforce aspect about the effect of dietary supplementation with HBP and SFE on immune response and fillet´s quality with the relationship the gene expression in specific tissues.

Authors: We modified the discussion regarding the immune response, despite the fact that there is little data in the literature on the relationship between quality, immunity and gene expression, which encouraged collaboration between the groups involved.

I believe that the manuscript needs major revision by the authors before being published in this prestigious journal

Reviewer 3 Report

Manuscript: Animals

Title: Effects of dietary supplementation with honeybee pollen and its supercritical fluid extract on immune response and fillet's quality of farmed gilthead seabream (Sparus aurata)

Overall comment:

The aim of this study was to analyse the potential of honey honeybee pollen and its supercritical fluid extract as a dietary supplement to promote fish welfare, by enhancing fish immune response, and simultaneously to provide seabream fillet a composition to better handle freezing conditions. Both themes are important and actual areas of research, aligned with Europe research agenda to promote fish health and provide consumers with high quality seafood products.

Manuscript is well organized, structured and clearly explained most of the times. Some parts might benefit to include more information (e.g. Introduction). On the Material and Methods, brief information is provided about the rearing. The analytical methods are adequate, but maybe insufficient for an adequate characterization of fish immune response. The techniques use to described fillets quality after storage are the adequate. The statistical analysis is also adequate, but from the figures is not clear what kind of comparison was done among treatments due to the quantity of superscripts used.

Some drawbacks hinder the manuscript quality. First, scarce information about fish performance. After a 30 days trial with 4 treatments and a control, authors just provide a mean value of the weigh. More details are needed on zoo-technical indexes (SGR, FCR), since these are important to evaluate fish health condition, which cannot be dissociated from immune response. This information can be provided as supplementary material. Fish trial and immune response seem to have a secondary role on this manuscript when compared with the weight of fillet characterization, which might reflect authors more background on this last area. Discussion can be improved

Not clear why authors decide to present polen and SFE  composition, since it does not seem relevant for this study. Incorporation on the diet is quite low, it would be more important to assess the FA of the diet. Considering the percentage of polen that was used on the diet, what is the quantity in the diet and expected to have been ingested by seabream? Another aspect is that maybe authors should have more caution when withdrawing conclusions on of the benefits of polen as a bioactive compound on immune system, a very complex system, just based on IL’s expression. Specially, when just treatment E1 was significantly different from control.

Based on the above mentioned I considered that manuscript needs major revisions to meet the quality standards to be published on ANIMALS

Specific comments and suggestions:

Line 137 - Simplify this sentence: “…conditions, for adaptation to the new conditions”

Line 129 – Suggestion: “Furthermore, water quality parameters were ….

Line 142 – Suggestion: Normally Feed

Line 233 – Not sure if authors can consider treatment a level, normally it is referred factor lipid, at 5 levels when testing 0, 20, 30, 40, 60 80 %; can treatment be considered a factor since two different forms of polen is used?

Line 236 – Please us identified instead of separate

Line 245 – Problems with the error bars in some of the treatments, since positive and negative have a different size.

Line 308 – It is not clear why authors need to describe fatty acid composition of HBP and HBP_SFE. This is the characterization of the product that was incorporated at a low percentage in the feed formulations. So, for this study what should be important was to characterize the experimental diets instead of the ingredient itself. During feed fabrication there substantial losses. Do authors have info on the  FA composition of the diet?

Line 341 – Figure 2 - Legends of the figures must indicate what the plot is describing; it seems it starts with the statistical analysis used. The number of the figures is wrong.

Line 348 -  Again Figure 1 – and the same comment  on Line 341

Line 370 – Suggestion since authors are discussion IL’s maybe can continue in the same paragraph.  What is IL-6 involved in? Some explanations were provided for IL1 & IL 8.

Line 402 – Maybe authors could conclude something with their own results.

Line 403-404 - This sentence is not clear. It seems something is missing.

Line 441 – Maybe delaying instead of delay

Line 450-435 – What does this mean? Is it a benefit?

Author Response

Reviewer 3

Overall comment:

R#3 The aim of this study was to analyse the potential of honey honeybee pollen and its supercritical fluid extract as a dietary supplement to promote fish welfare, by enhancing fish immune response, and simultaneously to provide seabream fillet a composition to better handle freezing conditions. Both themes are important and actual areas of research, aligned with Europe research agenda to promote fish health and provide consumers with high quality seafood products. Manuscript is well organized, structured and clearly explained most of the times. Some parts might benefit to include more information (e.g. Introduction). On the Material and Methods, brief information is provided about the rearing. The analytical methods are adequate, but maybe insufficient for an adequate characterization of fish immune response. The techniques use to described fillets quality after storage are the adequate. The statistical analysis is also adequate, but from the figures is not clear what kind of comparison was done among treatments due to the quantity of superscripts used.

Authors: We are grateful to R#3 for the positive evaluation. Caption of Figures have been modified to clarify the comparison done.

R#3 Some drawbacks hinder the manuscript quality. First, scarce information about fish performance. After a 30 days trial with 4 treatments and a control, authors just provide a mean value of the weigh. More details are needed on zoo-technical indexes (SGR, FCR), since these are important to evaluate fish health condition, which cannot be dissociated from immune response. This information can be provided as supplementary material. Fish trial and immune response seem to have a secondary role on this manuscript when compared with the weight of fillet characterization, which might reflect authors more background on this last area. Discussion can be improved

Authors: This trial is part of a wider series of studies aimed to test the possible effects of honeybee pollen on several aspects of fish growth, immune response, fish quality etc. In particular, in a first trial we focused on how the use of different inclusion levels of raw HBP in the diet can influence growth, nutrients digestibility, intestine histology, immunohistochemistry, hepatic biomolecular markers and blood biochemical analyses in a carnivorous fish species (Panettieri et al. 2020, cit.). In this trial instead we preferred to focus our research on immune response and organoleptic quality of the fillets rather than on growth performance. However, thank you for your observation. A table was added as supplementary material showing the main growth performance of the fish that did not show significant differences among groups, even if the diet with a greater inclusion of raw pollen resulted in tendentially worsened performance, confirming what already observed in Panettieri et al. (2020). This was also reported in the text in the “Results” section and in the “Discussion” section that were implemented.

R#3 Not clear why authors decide to present pollen and SFE composition, since it does not seem relevant for this study. Incorporation on the diet is quite low, it would be more important to assess the FA of the diet. Considering the percentage of pollen that was used on the diet, what is the quantity in the diet and expected to have been ingested by seabream? Another aspect is that maybe authors should have more caution when withdrawing conclusions on of the benefits of polen as a bioactive compound on immune system, a very complex system, just based on IL’s expression. Specially, when just treatment E1 was significantly different from control. Based on the above mentioned I considered that manuscript needs major revisions to meet the quality standards to be published on ANIMALS

Authors: Dear Referee, thanks for your comment, in our recent paper we showed that diets supplemented with SFE bee pollen extract had a stimulatory effect on fish serum immunity, respect to the inclusion of raw pollen. In this experiment, we wanted to investigate aspects that had not been analyzed in the previous already published paper. For what concerns the quantity of pollen ingested by the fish, it can be easily calculated from the ingestion, and it is 0.9 g/kg of average body weight per day and 0.43 g/kg of average body weight per day for P10 and P5 respectively. The quantity of extract added to the diets, instead, was determined to have the same total polyphenols content both in diets P5 and P10 and in diets E0.5 and E1 according to HBP extraction yield and extract phenol content, as determined by Messina et al. (2020). The point was added to Discussion. The text has been deeply revised, from Introduction to Conclusion.

Specific comments and suggestions:

R#3 Line 137 - Simplify this sentence: “…conditions, for adaptation to the new conditions”

Authors: The sentence has been corrected.

R#3 Line 139 – Suggestion: “Furthermore, water quality parameters were ….

Authors: Thank you, the sentence has been added.

R#3 Line 142 – Suggestion: Normally Feed

Authors: Thank you, we have corrected the sentence.

R#3 Line 233 – Not sure if authors can consider treatment a level, normally it is referred factor lipid, at 5 levels when testing 0, 20, 30, 40, 60 80 %; can treatment be considered a factor since two different forms of polen is used?

Authors: Thank you for the specification. We agree with you, here the diets cannot be considered as “levels” hence, we have modified the sentence.

R#3 Line 236 – Please us identified instead of separate

Authors: “Identified” has been utilized instead of separated. Thank you.

R#3 Line 245 – Problems with the error bars in some of the treatments, since positive and negative have a different size.

Authors: Thanks for the suggestion, we have changed the title of the y-axis to show that the we used a logarithmic scale, for this reason the error bars since positive and negative have a different size.

R#3 Line 308 – It is not clear why authors need to describe fatty acid composition of HBP and HBP_SFE. This is the characterization of the product that was incorporated at a low percentage in the feed formulations. So, for this study what should be important was to characterize the experimental diets instead of the ingredient itself. During feed fabrication there substantial losses. Do authors have info on the FA composition of the diet?

Authors: Thank you. We have moved the FA composition of HPB and SFE to the Supplementary material (Table S1). We agree with R#3, however we cannot provide the FA profile of the diets for this trial due to a material mistake. For this reason, we have shortened the discussion about FA and given minor emphasis to this parameter. We are sorry for this inconvenience.

R#3 Line 341 – Figure 2 - Legends of the figures must indicate what the plot is describing; it seems it starts with the statistical analysis used. The number of the figures is wrong.

Authors: The captions of Fig. 2 and 3 have been modified. We hope that now it is clearer than before.

R#3 Line 348 -Again Figure 1 – and the same comment on Line 341.

Authors: Thank you, please note the previous answer.

R#3 Line 370 – Suggestion since authors are discussion IL’s maybe can continue in the same paragraph. What is IL-6 involved in? Some explanations were provided for IL1 & IL 8.

Authors: Thank you for your suggestion. We provided the explanations for il-6.

R#3 Line 402 – Maybe authors could conclude something with their own results.

R#3 Line 403-404 - This sentence is not clear. It seems something is missing.

Authors: Please, considered the revised version of Discussion. It has been adjusted according to the suggestion of all of you.

R#3 Line 441 – Maybe delaying instead of delay.

Authors: The correction has been done.

R#3 Line 450-435 – What does this mean? Is it a benefit?

Authors: Thank you for your comment. The discussion is completely modified, please consider the new version.

The changes have been done accordingly to the all referee’ suggestions and we hope that the revised manuscript resulted now sufficiently improved to deserve the publication.

Thanks in advance to the precious work made by the Editor and by the three anonymous referees.

Round 2

Reviewer 1 Report

the manuscript has been successfully revised.